# Reinforcement Learning for Control with Stability Guarantee

## Abstract

Reinforcement learning (RL) has achieved promising performance in complicated system control. However, current RL-based control systems cannot utilize advanced RL methods while considering stability guarantees. To overcome this defect, we firstly apply a Lyapunov stable dynamical model as a reference system to fit the real system. Then, we prove that if the state fitting error between the reference and real system are bounded, the real system has Uniformly Ultimately Bounded (UUB) stability guarantee. Motivated by our theoretical analysis, we guide the design of reward functions for RL based on conditions of UUB guarantee for real systems and propose **ITSRL**, an **I**terative **T**raining framework for learning **S**table **RL** control policy with UUB stability guarantee, by iteratively minimizing the state fitting error between the reference and real system, which can be adapted to various advanced RL methods. Our evaluation results on three control tasks demonstrate that the proposed ITSRL framework can improve the performance of RL controller under perturbation.

## 1 Introduction

Modern control theory typically requires the understanding of system dynamics in order to derive the optimal controller (Shinners, Shinners (1998)). However, for complicated non-linear systems where accurately modeling the system dynamics is challenging, it is difficult for modern control theory to find the optimal controller (Hangos et al., 2004). Reinforcement learning (RL), due to its model-free nature, has been used to solve optimal control problems with unknown system dynamics, such as robotic control, unmanned autopilot control, etc. (Wiering & Van Otterlo, 2012).

However, existing RL approaches rarely consider the stability guarantee of the control system (Buşoniu et al., 2018), which is the most fundamental property for any control system (Sastry, 2013). Adaptive Dynamic Programming (ADP, Yang et al., 2016) design the controller with Uniformly Ultimately Bounded (UUB, Jain & Bhasin, 2017) stability guarantee, but it requires the model structure as a prior. Lyapunov Actor-Critic (LAC, Han et al., 2020) and Lyapunov-based soft actor–critic(LSAC, Han et al., 2021) learn the control policy which is stable in mean cost and UUB. Both methods are designed based on specific RL methods to meet the stability guarantee, which cannot be generalized to other advanced RL methods. However, different systems require different methods to achieve SOTA performance, so there are still challenges in applying the above methods to control systems.

In this paper, we propose a novel and general approach for stability analysis of real systems with control policy, which does not rely any assumptions about Markov chain induced by RL control policy. Specifically, we prove that if the state fitting error between a real system and the reference system can be bounded, then the real system has UUB stability guarantee. Motivated by our theoretical results, we guide the design of reward functions for RL based on conditions of UUB guarantee for real systems and establish **ITSRL**, an **I**terative **T**raining framework that can learn **S**table **RL** control policy by minimizing the state fitting error between a real system and the reference system. For ITSRL, RL methods are equivalent to plug-ins in the framework, and we can adopt any advanced RL method to adapt different control systems while the stability guarantee of the control policy can be satisfied as well. Specifically, we start by learning a stable dynamical model as a reference system of the real system. Then, instead of only using the original control objective as reward when optimizing RL control policy, we include the state fitting error as an additional penalty in the reward. By iterating these two steps, the performance of the RL control policy is improved as the fitting error decreases.To demonstrate the effectiveness of the proposed ITSRL framework, we evaluate it

on three control tasks with external input disturbance. Our evaluation results show that the proposed ITSRL can improve the performance of RL control policy under perturbation, compared with RL control policy without using our frameworks.

The contributions of this paper can be summarized as follows:

- We design an exponentially stable reference system with controller by applying the learning-based stable dynamic system model. We prove that the UUB stability of the real system can be guaranteed, if the state fitting error between a real system and the reference system can be bounded

- We propose an iterative training framework named ITSRL to train a RL control policy with UUB stability guarantee by minizing the state fitting error between the real system and the reference system.

- Experiments on multiple control tasks demonstrate that the proposed ITSRL can improve the performance of RL control policy under external perturbations.

## 2 RELATED WORK

Incorporating machine learning to design controllers has made great strides in recent years. Watter et al., 2015 introduces the embed to control (E2C) to learn non-linear dynamics with control from high-dimensional observations. Zhong et al., 2019 introduces the symplectic network deep learning framework to learn the Hamiltonian dynamic model with control, which takes physics-based priors into consideration. Nevertheless, these methods feed the control input directly into the neural network as the training dataset, without taking the design of the controller into account. Reinforcement learning (RL, Sutton et al., 1992) evaluates and optimizes policies through the interaction of agents with environment, which has shown stunning results in solving nonlinear control problems (Kaufmann et al., 2023).

However, machine learning-based control controllers have limitations in their applications as they cannot ensure stability on the controlled system (Sun, 2015). Therefore, it is natural to combine control theory and machine learning to develop learned control strategies with stability guarantees (Buşoniu et al., 2018). Lederer et al. (2021) propose a Gaussian process-based approach to quantify the impact of data on control performance and develop a control law that ensures the stability and performance of a closed-loop system despite model uncertainty. However, this uncertainty does not consider the disparities between the fitted system and the real system. Consequently, the learned control law may not perform well in the real system. Reinforcement learning is classified into two types of model-based (Moerland et al., 2023) and policy-based (Sewak & Sewak, 2019) learning strategies, based on whether the system model is known or not. Adaptive Dynamic Programming (ADP) gradually approximates the solution of dynamic programming as it learns (Luo et al., 2019). This is achieved by modeling the system's dynamics and reward function to predict the outcome of different decisions. The designed controller ensures the UUB of the state and the estimation error in the uncertain system. However, ADP faces challenges in dealing with complex environments that can hardly be modeled since it needs to know the model structure of the system, which is the common bottle-neck of model-based reinforcement learning as well. Han et al., 2020 proposes a data-driven UUB theorem without the prior knowledge of the system model, and develops Lyapunov-based actor-critic algorithm (LAC) to learn controllers with UUB guarantees. Based on LAC, Han et al., 2021 designs LSAC, which extends LAC into Markov Decision Process with safety constraints. However, both of these works assume that the Markov chain induced by the RL control policy is ergodic with a unique stationary distribution, which may not hold for some practical systems. By contrast, our stability guarantee is more general, without relying on any assumptions of the Markov chain induced by the RL control policy.

## 3 PRELIMINARIES

In this paper we mainly consider the case of dynamics control system $\dot{x} = f(x(t), u(x(t))$ for $x(t) \in \mathbb{R}^n$, $u(x) \in \mathbb{R}^m$. It is necessary to introduce the stability in the sense of Lyapunov, which is the basis for further determining the subsequent gradual stability and exponential stability.

**Lyapunov Function.** Lyapunov second method formulate the Lyapunov function that can estimate the stability of the system based on the theory of energy in physics. For dynamical system, if a continuously differentiable function $V(x,t)$ can be constructed with all non-zero states in the state space satisfy: (i) $V(x,t)$ is positive definite and bounded, i.e. there exists two continuously non-decreasing functions $a_1(\|x\|)$ and $a_2(\|x\|)$, where $a_1(0) = a_2(0) = 0$, such that the inequality below

$$a_1(\|x\|) \leq V(x,t) \leq a_2(\|x\|) \tag{1}$$

is satisfied for $\forall t \in [t_0, +\infty], x \neq 0$ (ii) The derivative of $V(x,t)$ with respect to time $t$ is negative definite and bounded, which means there exists a continuously non-decreasing function $a_3(\|x\|)$, $a_3(0) = 0$, such that the inequality below

$$\dot{V}(x,t) \leq -a_3(\|x\|\|) \tag{2}$$

is satisfied for $\forall t \in [t_0, +\infty], x \neq 0$

**Input to State Stability.** Consider the system of ordinary differential equations with external input(for control system the external input here is pointed out to be system errors, external disturbances, etc.)

$$\dot{x} = f(x, \omega), x(0) = x_0, \tag{3}$$

where $x(t) \in \mathbb{R}^n$, $\omega \in \mathbb{R}^m$ is bounded in terms of infinite norm, the affine $f : \mathbb{R}^n \times \mathbb{R}^m \to \mathbb{R}^n$, $f(0,0) = 0$ is locally Lipschitz. The system (3) above is called to be input-to-state stable (ISS, Agrachev et al., 2008) if there exist $a_1 \in \mathcal{KL}$ and $a_2 \in \mathcal{K}$ such that for any initial value $x_0 \in \mathbb{R}^n$ and any external input $\omega \in \mathbb{R}^m$ the corresponding solution $x = \phi(\cdot, x_0, \omega)$ exists on $[0.\infty)$ and satisfies

$$|\phi(t, x_0, \omega)| \leq a_1(|x_0|, t) + a_2(\|u\|_\infty) \tag{4}$$

The proposition follow gives the definition of an ISS-Lyapunov function.

**Proposition 1.** *A smooth function V is an ISS-Lyapunov function for (3) if and only if there exist $\alpha_i \in \mathcal{K}\infty(1 \leq i \leq 4)$ such that*

$$\alpha_1(\|x\|) \leq V(t, x) \leq \alpha_2(\|x\|) \tag{5}$$

$$\nabla V(x)f(x, \omega) \leq -\alpha_3(\|x\|) + \alpha_4(\|\omega\|) \tag{6}$$

**Uniformly Ultimately Bounded.** Uniformly ultimately bounded (UUB) is also an approach to characterize the stability of uncertain systems. The UUB starts from the initial value of the system and considers the final stability region of the system, which is a degree of proximity to the final state to the origin equilibrium point of the uncertain system.

Consider the system $\dot{x} = f(t, x)$, where $f : [0, \infty) \times D \to \mathbb{R}^n$ is piecewise continuous in $t$, locally Lipschitz in $x$ on $[0, \infty) \times D$, and $D \subset \mathbb{R}^n$ is a domain that contains the origin $x_e = 0$. The solutions of system above is uniformly ultimately bounded if there exists positive constants $b$ and $c$, independent of $t_0 \geq 0$, and for $\forall a \in (0, c)$, there is $T = T(a, b)$, independent of $t_0$, such that

$$\|x(t_0)\| \leq a \Rightarrow \|x(t)\| \leq b, \quad \forall t \geq t_0 + T \tag{7}$$

## 4 REINFORCEMENT LEARNING FOR CONTROL WITH STABILITY GUARANTEE

In this section, the design of the stable reference system is firstly given. Then the theoretical analysis of the stability of the real system with the controller is provided, by assuming that the state fitting error between the stable reference system and the real system is bounded. Finally, the design of the proposed ITSRL framework that can learn stable RL control policy with UUB guarantee is described.

Given a real system $\dot{x} = f(x, u(x))$ with control policy $u(\cdot)$, we can learn a stable dynamic system $\dot{\hat{x}} = \hat{f}(\hat{x}, u(\hat{x}))$ to fit the aforementioned real system and then use the learnt stable as a reference system. If the fitting error between the real system and the reference system converges to zero, the real system can also be stable. With the consistent initial state $x_0$ and the controller $u(\cdot)$, we can obtain the state sequence and furthermore the error sequence from both system, where the error $e$

between the reference and the real system can be directly observed from states or implicitly from system models. It is worth noting that although errors may be observed between both system models, modeling the real system is unnecessary. Thanks to the data-driven reference system used to fit the real environment, we solely need to analyze the impact of error bounds on the stability analysis of the real system. Finally as for the two cases of observed errors, we separately provide the ISS analysis when the error is observed from states, and the UUB analysis when the error is implicitly observed from system models, for the real system.

## 4.1 THE STABLE REFERENCE SYSTEM

Following the framework mentioned above, it is significant to provide a data-driven reference system with the stability guarantee. Let $\widetilde{f} : \mathbb{R}^n \to \mathbb{R}^n$ denote the common dynamics model composed of multi-layer perceptron (MLP). The design of reference system is defined as:

$$
\begin{aligned}
\hat{f}(\hat{x}, u) &= \textbf{Proj}(\widetilde{f}(\hat{x}, u), \{\hat{f} : \nabla V(\hat{x})\hat{f} \leq -\alpha V(\hat{x})\}) \\
&= \begin{cases} \widetilde{f}(\hat{x}, u) & \text{if } \nabla V(\hat{x})\widetilde{f} \leq -\alpha V(\hat{x}) \\ \widetilde{f}(\hat{x}, u) - \nabla V(\hat{x})\frac{\nabla V(\hat{x})\widetilde{f}(\hat{x},u)+\alpha V(\hat{x})}{\|V(\hat{x})\|_2^2} & \text{otherwise} \end{cases} \\
&= \widetilde{f}(\hat{x}, u) - \nabla V(\hat{x})\frac{\text{ReLU}(\nabla V(\hat{x})\hat{f}(\hat{x}, u) + \alpha V(\hat{x}))}{\|V(\hat{x})\|_2^2}
\end{aligned}
\tag{8}
$$

where $\textbf{Proj}(A, B)$ denotes the orthogonal projection of $A$ onto $B$. The second equation comes from the analytical projection of a point onto the half-space. The Lyapunov function $V : \mathbb{R}^n \to \mathbb{R}$ is designed as below.

$$
\begin{aligned}
z_1 &= \sigma_0(W_0\hat{x} + b_0) \\
z_{i+1} &= \sigma_i(U_i z_i + W_i\hat{x} + b_i) \\
g(\hat{x}) &\equiv z_k \\
V(\hat{x}) &= \sigma_{k+1}(g(\hat{x}) - g(0)) + \epsilon\|\hat{x}\|_2^2
\end{aligned}
\tag{9}
$$

where the quadratic regularization term $\epsilon\|\hat{x}\|_2^2$ makes sure the strict positive definiteness of $V$, the function $g(\hat{x})$ is the ICNN (Amos et al., 2017) with real-value weights $W_i$, positive weights $U_i$ and real valued bias $b_i$, $\sigma_i(\hat{x})(i = 0, 1, 2\dots)$ are nonlinear convex, non-decreasing monotonically smoothed ReLU activations with a quadratic region in $[0, d], d \in (0, 1)$.

$$
\sigma_i(\hat{x}) = \begin{cases} 0 & \hat{x} \leq 0 \\ \dfrac{\hat{x}^2}{2d} & 0 < \hat{x} < d \\ \hat{x} - \dfrac{d}{2} & \hat{x} \geq d \end{cases}
\tag{10}
$$

## 4.2 STABILITY ANALYSIS

For the reference system defined above, we can draw the following inference referred in Kolter & Manek, 2019.

**Lemma 1.** *The dynamical control systems*

$$
\dot{\hat{x}} = \hat{f}(\hat{x}, u)
\tag{11}
$$

*defined by $\hat{f}$ from (8) and $V$ from (9) are globally exponentially stable to the equilibrium point $\hat{x} = 0$, for any (bounded weight) networks defining the $\widetilde{f}$ and $V$ function.*

With the stability of the reference system, the ISS and the UUB analysis to the real system can be respectively proposed under different conditions. Firstly, we make the following assumption:

**Assumption 1.** *The error over time $e(t)$ between the reference and the real system is bounded and Lipschitz continuous with $e(0) = 0$.*

which is a common assumption for the optimal control. The assumption $e(0) = 0$ is hold since the initial state for both systems is the same. With the assumption above, we can obtain the following analysis as the error is observed under respective conditions.

**Input-to-State Stability Analysis.** If the error $e$ can be directly observed from states, the following stability analysis of the control system can be given.

**Theorem 1.** *Under Assumption 1, consider the real system under control*

$$\dot{x} = f(x, u(x)) \tag{12}$$

*The reference system with the controller $\dot{\hat{x}} = \hat{f}(\hat{x}, u(\hat{x}))$ is exponentially stable around $\hat{x}_e = 0$. If states between both systems satisfies*

$$x(t) = \hat{x}(t) + e(t), \quad \forall t > t_0 \tag{13}$$

*where $e(t)$ denotes the error observed from states of the real system and the reference system. Then the real system (12) under control is ISS stable while the $V$ defined from (9) is the ISS-Lyapunov function of the system (12).*

*Proof.* Assume that the solutions of real system and reference system with the same initial $x_0$ are $x = \phi(t, x_0, u)$ and $\hat{x} = \hat{\phi}(t, x_0, u)$. As $x = \hat{x} + e$ we have $\dot{x} = dx/dt = d(\hat{x} + e)/dt = \hat{f} + \dot{e}$. According to **Assumption 1**, let $e_{sup}^+$ as the positive upper bound of the error $e$ and let $L_e$ be the Lipschitz constant of $\dot{e}$. The definition of $\hat{f}$ implies that

$$\dot{V}(\hat{x}) = \nabla V(\hat{x})^T \hat{f}(\hat{x}, u(\hat{x})) \leq -\alpha V(\hat{x}(t)) \tag{14}$$

Based on (9), it can be obtained that $V(\hat{x})$ is a polynomial function composed of power terms of $\hat{x}$ and so is its derivative, thus $\nabla V(\hat{x} + e)$ can be written as

$$\begin{aligned} V(x) &= V(\hat{x} + e) = V(\hat{x}) + \mathbf{p}(\hat{x}, e|\ U_i, W_j, b_m) \\ \nabla V(x) &= \nabla V(\hat{x} + e) = \nabla V(\hat{x}) + \mathbf{p}_d(\hat{x}, e|\ U_i, W_j, b_m) \end{aligned} \tag{15}$$

where $\mathbf{p}(\hat{x}, e|\ U_i, W_j, b_m)$ and $\mathbf{p}_d(\hat{x}, e|\ U_i, W_j, b_m)$ are polynomial functions formulated by the power terms of $\hat{x}$ and $e$ with coefficient composed of the weight $U_i \in \mathbb{R}^+$, $W_j \in \mathbb{R}$, $b_m \in \mathbb{R}$, $(i, j, m = 0, 1, 2, \ldots k)$. Since the reference system is exponentially stable, it can be inferred that $x$ is bounded, define $\hat{x}_{sup}^+$ as the positive upper bound of $\hat{x}$. Thus we can conclude that polynomial functions $\mathbf{p}(\hat{x}, e|\ U_i, W_j, b_m)$ and $\nabla V(\hat{x}) + \mathbf{p}_d(\hat{x}, e|\ U_i, W_j, b_m)$ are both bounded as the error $e$ is bounded.

Moreover consider the neural network of reference system $\hat{x}_{k+1} = f(\hat{x}_k, u(\hat{x}_k))$, it can be obtained that

$$\lim_{k \to \infty} f(\hat{x}_k, u(\hat{x}_k)) = \lim_{k \to \infty} \hat{x}_{k+1} = 0 \tag{16}$$

which means that $f$ is upper bounded by some positive $f_{sup}^+$. Thus it can be inferred that

$$\begin{aligned} \dot{V}(x) &= \dot{V}(\hat{x} + e) = \nabla V(\hat{x} + e) \frac{d}{dt}(\hat{x} + e) \\ &= (\nabla V(\hat{x}) + \mathbf{p}_d(\hat{x}, e|\ U_i, W_j, b_m))(\hat{f} + \dot{e}) \\ &= \nabla V(\hat{x}) f + \mathbf{p}_d(\hat{x}, e|\ U_i, W_j, b_m) \hat{f} + \nabla V(x) \dot{e} \\ &\leq -\alpha V(\hat{x}) + f_{sup}^+ \mathbf{p}_d(e|\ x_{sup}^+, U_i, \|W_j\|, \|b_m\|) + \|\nabla V(x)\| L_e \\ &= -\alpha V(x) + \alpha \mathbf{p}(\hat{x}, e|\ U_i, W_j, b_m) + f_{sup}^+ \mathbf{p}_d(e|\ x_{sup}^+, U_i, \|W_j\|, \|b_m\|) + \|\nabla V(x)\| L_e \\ &\leq -\alpha V(x) + f_{sup}^+ \mathbf{p}_d(e|\ x_{sup}^+, U_i, \|W_j\|, \|b_m\|) + (\alpha \mathbf{p}_{sup}^+ + \|\nabla V(x)\| L_e) \end{aligned} \tag{17}$$

where $\mathbf{p}_{sup}^+$ is the positive upper bound of $\mathbf{p}(\hat{x}, e|\ U_i, W_j, b_m)$. It can be inferred that $x$ is bounded cause $\hat{x}$ and $e$ are both bounded, which can further obtain that the polynomial function $\nabla V(x)$ is bounded. Let $\mathcal{V}^+$ denote the positive upper bound of $\nabla V(x)$. Then we have:

$$\alpha \mathbf{p}_{sup}^+ + \|\nabla V(x)\| L_e \leq \alpha \mathbf{p}_{sup}^+ + \mathcal{V}^+ L_e \tag{18}$$

Let $\mathcal{N} = \alpha \mathbf{p}_{sup}^+ + \mathcal{V}^+ L_e$. By the definition of (9) we have $\epsilon \|x\|_2^2 \leq V(x)$, where the lower bound is decided by $\sigma_i(\cdot) \geq 0$ and $g$ is positive. Define $g(e) = f_{sup}^+ \mathbf{p}_d(e|\ x_{sup}^+, U_i, \|W_j\|, \|b_m\|) + \mathcal{N}$ and (17) can be written as:

$$\dot{V}(x) \leq -\alpha \epsilon \|x\|_2^2 + g(e) \tag{19}$$

Meanwhile by the definition of (9), we have $V(x) \leq M\|x\|_2^2$ based on the fact that $V(x)$ behaves linearly as $\|x\| \to \infty$ and is quadratic around the origin. Since $\epsilon\|x\|_2^2$, $M\|x\|_2^2$, $\alpha\epsilon\|x\|_2^2$ and $g(e)$ are class $\mathcal{K}\infty$ functions with respect to $x$ and $e$, which satisfies the **Proposition 1**. The $V$ defined from (9) is proved as the ISS-Lyapunov function to the control system. $\qquad\square$

**Uniformly Ultimate Boundedness stability Analysis.** If the error is observed from system models and accumulates as the system evolves. Consider the globally exponentially stability of the reference system, we can regard it as a "nominal" system so that the controller under the real system is equivalent to the perturbed "nominal" system, where the perturbation corresponds to the observed system model error. Then we can obtain the following theorem:

**Theorem 2.** *Consider the controller under the real system*

$$\dot{x} = f(x, u(x)) + e(t, x) \tag{20}$$

*where the reference system $\dot{x} = f(x, u(x))$ is globally exponentially stable around $\hat{x}_e = 0$, $e(t, x)$ is the observed system model error. Let $V(t, x)$ defined from (9) be the Lyapunov function of the reference that satisfies:*

$$\alpha_1(\|x\|) \leq V(t, x) \leq \alpha_2(\|x\|) \tag{21}$$

$$\dot{V}(t, x) = \frac{\partial V}{\partial t} + \frac{\partial V}{\partial x} f(t, x) \leq -\alpha_3(\|x\|) \tag{22}$$

$$\left\|\frac{\partial V}{\partial x}\right\| \leq \alpha_4(\|x\|) \tag{23}$$

*in $[0, \infty) \times D$, where $D = \{x \in \mathbb{R}^n | \|x\| < \mu\}$, where $\mu$ is the upper bound of the state $x$, the $\alpha_i(\|x\|)(i = 1, 2, 3, 4)$ are class $\mathcal{K}$ functions shown as below:*

$$\alpha_1(\|x\|) = \epsilon\|x\|^2, \ \alpha_2(\|x\|) = M\|x\|^2,$$
$$\alpha_3(\|x\|) = \alpha\epsilon\|x\|^2, \ \alpha_4(\|x\|) = \boldsymbol{p}_d(\|x\|_2 \mid U_i, \|W_j\|, \|b_m\|) \tag{24}$$

*Suppose the error $e(t)$ and initial state $\|x_0\|$ satisfies:*

$$\|e(t, x)\| \leq \delta < \frac{\theta\alpha_3(\alpha_2^{-1}(\alpha_1(\mu)))}{\alpha_4(\mu)} \tag{25}$$

$$\|x_0\| < \alpha_2^{-1}(\alpha_1(\mu)) \tag{26}$$

*for $\forall t > 0, x \in D$, and some positive constant $\theta < 1$. The real system (20) will be uniformly ultimately bounded, that is, the state $x(t)$ of the real system satisfies:*

$$\|x(t)\| \leq \beta(\|x(t_0)\|, t - t_0), \ \forall t_0 \leq t < t_0 + T \tag{27}$$

$$\|x(t)\| \leq \rho(\delta), \ \forall t \geq t_0 + T \tag{28}$$

*for some class $\mathcal{KL}$ function $\beta$ and some finite $T$ with the initial time $t_0$, where $\rho$ is a class $\mathcal{K}$ function of $\delta$ defined by*

$$\rho(\delta) = \alpha_1^{-1}\left(\alpha_2\left(\alpha_3^{-1}\left(\frac{\delta\alpha_4(r)}{\theta}\right)\right)\right) \tag{29}$$

*Proof.* According to the definition of the $V$ we have:

$$\epsilon\|x\|^2 \leq V(x) \leq M\|x\|^2 \tag{30}$$

where the lower bound follows by definition and the fact that $g$ is positive. The upper bound determined by the fact that the activation function $\sigma_i$ is defined to be linear when $x > d$ and quadratic around the origin, which result in that $V(x)$ behaves linearly as $\|x\| \to \infty$. Therefore $V(x)$ can be upper bounded by some quadratic term $M\|x\|^2$. Moreover $\dot{V}$ can be bounded on the basis of construction of $V$ and (30) we can obtain that:

$$\dot{V}(x, t) \leq -\alpha V(x) \leq -\alpha\epsilon\|x\|_2^2 \tag{31}$$

As for the gradient of $V$ with respect to $x$, we can obtain from the form of ICNN of $V$ that $\|\partial V/\partial x\|$ is a polynomial function $\boldsymbol{p}_d$ consisting of power terms of $x$ as follows

$$\left\|\frac{\partial V}{\partial x}\right\| = \boldsymbol{p}_d(\|x\||U_i, W_j, b_m) \tag{32}$$

where $U_i \in \mathbb{R}^+$, $W_j \in \mathbb{R}$, $b_m \in \mathbb{R}$ are the weights and bias in $V$. We can further get

$$\left\| \frac{\partial V}{\partial x} \right\| = \mathbf{p}_d(\|x\| \| U_i, W_j, b_m) \leq \mathbf{p}_d(\|x\| \| U_i, \|W_j\|, \|b_m\|) \tag{33}$$

It is straightforward to recognize that the polynomial function consisting of power terms of $\|x\|$ with all the positive real-value weights is a $\mathcal{K}$ function. Meanwhile it is evidence that $\epsilon\|x\|^2$, $M\|x\|^2$, $\alpha\epsilon\|x\|^2$ are $\mathcal{K}$ function of $\|x\|$. Thus we can infer that the Lyapunov function defined in (9) satisfies (21) through (23). We use $V(t, x)$ as a Lyapunov function for the real system. The derivative of $V(t, x)$ along the trajectories of the real system satisfies:

$$\begin{aligned}
\dot{V}(t, x) &\leq -\alpha_3(\|x\|) + \left\| \frac{\partial V}{\partial x} \right\| \|e(t, x)\| \\
&\leq -\alpha_3(\|x\|) + \delta\alpha_4(\|x\|) \\
&\leq -(1-\theta)\alpha_3(\|x\|) - \theta\alpha_3(\|x\|) + \delta\alpha_4(r), \ 0 < \theta < 1 \\
&\leq -(1-\theta)\alpha_3(\|x\|), \ \forall \|x\| \geq \alpha_3^{-1}\left(\frac{\delta\alpha_4(r)}{\theta}\right)
\end{aligned} \tag{34}$$

Then applying the lemma (Khalil, 2015, Page 172) can complete the proof. $\square$

### 4.3 ITERATIVE REINFORCEMENT LEARNING FRAMEWORK WITH STABILITY GUARANTEE

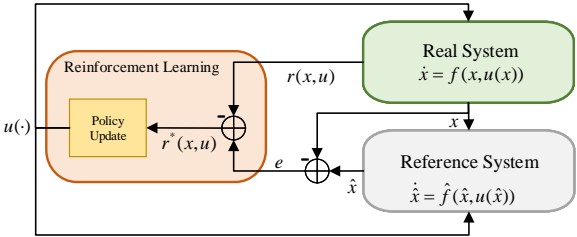

Figure 1: The proposed ITSRL framework learns the RL controller with stability guarantee by decreasing the observed error to satisfy the conditions of stability analysis.

Combined with the stability analysis in the previous subsection, we propose an iterative framework towards learning stable control system By minimizing the error between the reference system and the real system, the stability of the real system with the RL controller can be guaranteed. Note that we denote our framework by ITSRL in the rest of this paper. As shown in Figure 1, the RL controller $u_k$ will take the error as an additional feedback to enhance the performance of the RL controller. Specifically, instead of maximizing the reward function $r(x_k, u_k)$ of RL only, the RL controller will also be trained to minimize the fitting error $e$. Formally, the RL controller is trained with the reward $r^*$ below:

$$r^*(x_k, u_k) = r(x_k, u_k) - \lambda \|e\|_2, \tag{35}$$

where $\lambda$ is an adjustable parameter.

**Iterative Training Process.** The RL controller and the stable reference system model are trained iteratively. Specifically, given a pretrained RL controller, we firstly fix the RL controller and train the reference system model to fit the real system. Then, we fix the well-trained reference system model and train the RL controller to maximize the above reward function while minimizing the fitting error. We repeat these two processes until $r^*(x_k, u_k)$ converges.

## 5 EMPIRICAL RESULTS

In this section, we discuss the results related to the proposed ITSRL framework. Firstly, we empirically show that by only minimizing the fitting error between the reference system and the real system, the RL controller can be more stable, achieving higher reward. Our experimental results

are consistent with the theoretical analysis, which motivates the design of our proposed framework. Then we show the performance of the RL controller under the ITSRL framework, and compare it with the vanilla RL controller under different levels of perturbations. All experiments are conducted within OpenAI gym environments (Brockman et al., 2016) with Mujoco (Todorov et al., 2012), and we use the RL policy trained by PPO (Schulman et al., 2017) as our default RL controller.

## 5.1 MOTIVATION EXPERIMENT

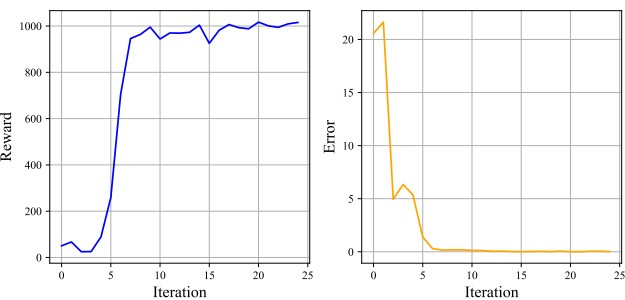

Figure 2: Hopper's training reward and fitting error during RL training by minimizing the fitting error only.

We experimentally demonstrate the effectiveness of the proposed iterative training framework, as shown in Figure 2. We eliminate the originally existing reward mechanism in the motivated experiments and only add the error between the real system (OpenAI gym) and the reference system as the reward evaluation term for the RL controller. As shown in Figure 2, by only using the error as a penalty term, the controller still gradually obtains better performance during the learning process. It is worth mentioning that the error between the two systems gradually decreases as the reward increases during the training process, which also demonstrates the optimization effect of our iterative training in terms of reducing the fitting error. This in turn allows the stability guarantee in Section 4 to be achieved.

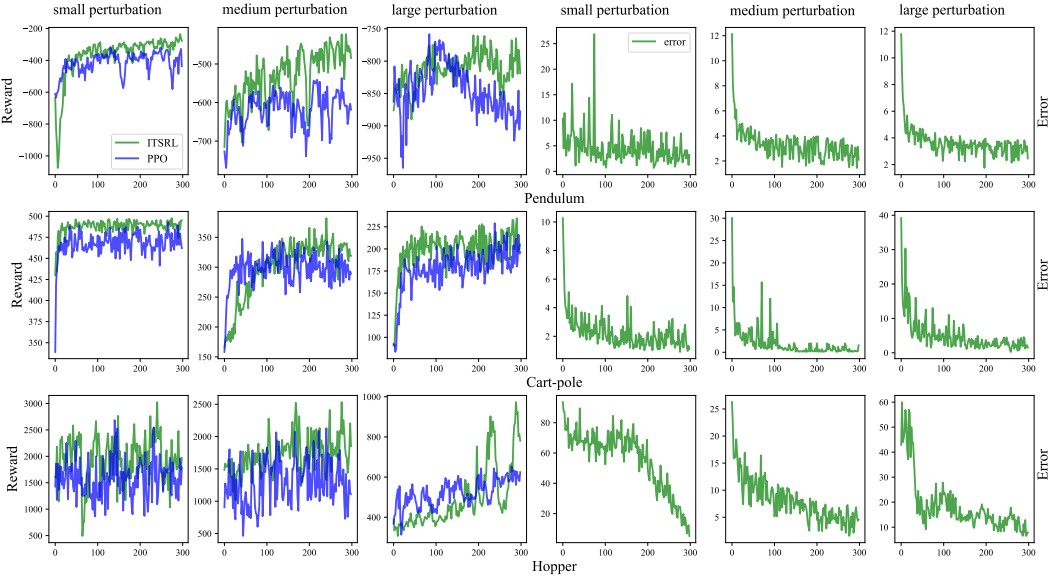

Figure 3: Results of anti-disturbance experiments. Note that the left three columns of figure show how the reward changes during training, under small, medium, and large perturbation. The right three columns of figures show how the state fitting error between the real system and the reference system changes during training, under small, medium, and large perturbation.

## 5.2 ANTI-DISTURBANCE EXPERIMENTS

We demonstrate the superiority of the ITSRL framework under different perturbations in different OpenAI gym environments. For anti-disturbance training, we firstly train agents with PPO with default parameters under each perturbation. The RL agents are then made more robust to the same perturbations by iterative training based on the ITSRL framework. As for reference system network, the Lyapunov function network and the common dynamics model network both have two hidden layers of sizes $(64, 64)$ with ReLU activation functions. The learning rate for common dynamics model is $3 \times 10^{-4}$, the learning rate of lyapunov function is $1 \times 10^{-4}$, and the Lyapunov constant is $0.9$. For evaluation, we test the performance and obtain the reward of these agents under a range of different perturbations subjected to no greater than the maximum case at training.

**Cart Pole.** In this experiment, we set the learning rate as $10^{-4}$. Since CartPole environment uses a binary value (i.e. $0$ and $1$) as the control input, we inject perturbation by randomly flipping the control input with probability $\sigma$. Specifically, we choose $\sigma$ from $\{0, 0.1, 0.2, 0.25\}$, corresponding to zero, small, medium, and large perturbations.

**Pendulum.** In this experiment, we set the learning rate as $4 \times 10^{-5}$. Considering that Pendulum environment uses a vector with real values as input, we inject perturbation by adding Gaussian random noise with zero mean and standard deviation of $\sigma$ to the control input. Specifically, we select $\sigma$ from $\{0, 1.5, 2, 3\}$, which are defined as zero, small, medium and large perturbations respectively.

**Hopper** In this experiment, we set the learning rate as $4 \times 10^{-5}$. The perturbation is added as the Gaussian random noise with with zero mean and standard deviation of $\sigma$ to the control input.Specifically, we select $\sigma$ from $\{0, 0.1, 0.5, 1\}$, which are defined as zero, small, medium and large perturbations respectively.

| Task | Zero Perturbation | | Small Perturbation | | Medium Perturbation | | Large Perturbation | |
|---|---|---|---|---|---|---|---|---|
| | PPO | ITSRL | PPO | ITSRL | PPO | ITSRL | PPO | ITSRL |
| Cartpole | 500.0 | **500.0** | 471.0 | **487.6** | 318.8 | **350.9** | 185.6 | **201.2** |
| Pendulum | -222.9 | **-221.2** | -386.6 | **-310.4** | -612.4 | **-480.8** | -904.6 | **-835.4** |
| Hopper | 2634.1 | **2889.3** | 2162.5 | **2794.2** | 1800.7 | **2015.8** | 634.6 | **1103.7** |

Table 1: Comparison of testing reward between PPO and ITSRL under different amount of perturbations on three control tasks.

Figure 3 and Table 1 show the improvement in the training reward and testing reward of the RL controller with and without the ITSRL framework under three levels of perturbation respectively. Through the testing results, we can observe that in the three groups of anti-disturbance experiments under each control task, the agent's performance can be improved. Meanwhile, Figure 3 present the changes of state fitting error during iterative learning process, which demonstrate that the error indeed converges during training process.

## 6 CONCLUSION

This paper first introduces a reference system with RL controller which has exponential stability property. Then it gives the stability analysis of the real system with RL controller, assuming that the fitting error between the reference system and the real system can be bounded. Furthermore, a novel iterative learning framework motivated by the stability analysis is developed to improve the stability performance of RL controller in the face of perturbations by reducing the state error. Future direction involves optimizing the ITSRL framework to further improve the robustness of RL controller and reduce the training time.

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
