# OpenReview forum: "Reinforcement Learning for Control with Stability Guarantee"
_ICLR.cc/2024/Conference — Submitted to ICLR 2024_

### Official Review · Reviewer_4p4H · 2023-10-29

**Soundness:** 2 fair
**Presentation:** 2 fair
**Contribution:** 3 good
**Rating:** 3
**Confidence:** 4

**Summary:**

This paper presents an Iterative Training framework for learning Stable RL control policy with a Uniformly Ultimately Bounded (UUB) stability guarantee. They first tailor the idea of learning a stable deep network model from [Kolter & Manek, 2019] to show that with an exponentially stable reference model, as long as the state error between the real system and reference model is bounded, the real system is guaranteed to be ISS stable (theorem 1) and UUB (theorem 2). With this observation, they propose to penalize the error in the training process and obtain some reasonable experiment results on three benchmark problems.

**Strengths:**

The RL for control with stability guarantee is nice, and the authors provide some theoretical analysis for this. I believe that the algorithm proposed by the authors is new and empirical results show that their methods outperform the vanilla RL controller under different levels of perturbations.

**Weaknesses:**

Although I really appreciate the theoretical analysis by the authors in Section 4. The reference model is a simple adoption from [Kolter & Manek, 2019] and the proofs are fairly standard in nonlinear control analysis. In addition, all the theoretical analysis is based on Assumption 1, bounded state error, which still cannot be theoretically guaranteed during the training process under ITSRL. Also, the pseudo-code for the main algorithm ITSRL is missing.

The presentation quality of this paper can be further improved. The notations are abused in several places, sometimes not properly defined. Some example:
1. it is important to first introduce the class of functions $\mathcal{K}$, $\mathcal{KL}$, and $\mathcal{K_\infty}$. These are standard in the control community but they are not that common in the learning field.
2. In section 3, $a_i$ functions are used in both the Lyapunov function and ISS paragraphs, but they are supposed to belong to different classes of functions.
3. For the system dynamics $f$, sometimes it appears as $f(x,u)$, sometimes becomes $f(x, w)$ and $f(x,t)$. I understand that these notations are served for different purposes. But this inconsistency can be misleading and it would be great if you could clarify their difference.
4. Missing notation $i$ in equation 9.
5. $g$ is used twice in the paper for different functions: equation 9 and $g(e)$ defined above equation 19.

**Questions:**

1. What do you mean $f(0,0)=0$ is locally Lipschitz under equation 3?
2. What is the reference for Proposition 1?
3. What do you mean by the common dynamics model in section 4.1?
4. Are $\sigma$ functions defined in equation 10 smooth enough? It is not continuously differentiable, do you require the Lyapunov function to be continuously differentiable in the theoretical analysis?
5. There are some typos in equation 17 (it should be $\hat{f}$ in the third equality and there is a missing term $\mathbf{p}_d \cdot \dot{e}$ in this line as well). Also, I do not follow the last equality. Please rewrite this equation carefully and explain the key steps.
6. Which lemma are you referring to from [Khalil 2015] for proving Theorem 2?

---

### Official Review · Reviewer_KFTS · 2023-10-29

**Soundness:** 2 fair
**Presentation:** 1 poor
**Contribution:** 2 fair
**Rating:** 1
**Confidence:** 3

**Summary:**

The paper proposed stability guarantees for RL algorithm with uniformly ultimate soundness stability analysis. The real system are projected on a data-driven reference system using learning-based methods. Theoretical results claim that if the error is Lipschitz and bounded, then the real system is also stable. Experimental results showed that the methods are robust to perturbations.

**Strengths:**

The idea is interesting. Cannot really find any other strengths in the quality, clarity, and significance.

**Weaknesses:**

1. Presentation is bad. There are a lot of unclear technical details, including but not limited to these:
   - No clear algorithm for Figure 1.
   - No clear definition of model error and reward in algorithms or experimental results.
   - Notations are abused like e(t) in Assumption 1 and e(t,x) in thoerem 2.
2. Assumptions are too strong. It is hard to say you can project one real system on a given network structure (it is even convex) with bounded error. No discussion on how realistic the assumptions are. There are also hidden assumptions that you can find a Lyapunov function V satisfying conditions in eq 9. No discussions on the difficulties of finding a legitimate Lyapunov function of a given system, which should be the core difficulties of Lyapunov-based methods.
3. Missing related works. The abstract claims that *current RL-based control systems cannot utilize advanced RL methods while considering stability guarantees*. However, there is rich literature discussing RL with stability/safety guarantees (they are very similar), such as
   - Berkenkamp, Felix, et al. "Safe model-based reinforcement learning with stability guarantees." *Advances in neural information processing systems* 30 (2017).
   - Cheng, Richard, et al. "End-to-end safe reinforcement learning through barrier functions for safety-critical continuous control tasks." *Proceedings of the AAAI conference on artificial intelligence*. Vol. 33. No. 01. 2019.
   - Choi, Jason, et al. "Reinforcement learning for safety-critical control under model uncertainty, using control lyapunov functions and control barrier functions." *arXiv preprint arXiv:2004.07584* (2020).
4. Experimental results are confusing. Settings are not clear. No evidence that either PPO or proposed algorithms are STABLE, given that stability is the major concern of this paper.
5. Nothing in the theoretical analysis is really related to reinforcement learning. The analysis is more like analyzing two dynamical systems. What if we cannot solve the optimal policy using RL? What if we cannot learn the accurate system from data? These problems are common in RL and should be at least discussed. I suggest submitting this paper to a control conference.

**Questions:**

1. The major technical questions: You mess up three types of systems,

   1. ISS stable systems

   2. Systems where a Lyapunov function V could be found to satisfy UUB

   3. Systems could be captured by (9) with bounded error

      Are they the same family of systems?

2. How do you evaluate the model learning error and how to learn the reference system in Figure 1? Please explain the algorithm details.

3. See Weakness 5.

---

### Official Review · Reviewer_F9dy · 2023-10-30

**Soundness:** 2 fair
**Presentation:** 3 good
**Contribution:** 3 good
**Rating:** 3
**Confidence:** 4

**Summary:**

The paper considers the problem of stability guarantees in RL. To address the problem, the paper introduces a reference system to fit the real system, and prove that if the state fitting error between the reference and the real system are bounded, the real system has UUB stability guarantee. Then, the paper proposes a novel method for designing reward functions for RL, aiming at minimizing the error between the real system and the reference system. Experiments are done on three environments to show that the proposed framework can improve the performance of RL.

**Strengths:**

The writing of the paper is clear and easy to follow. The problem considered is important. The proposed method for designing RL reward functions is novel and interesting.

**Weaknesses:**

1. The assumptions of the theoretical analysis are not fully justified. The authors are encouraged to give each assumption a justification to show that if this assumption is easy to be fulfilled in the real world. For example, in Assumption 1, it is assumed that $e(t)$ is bounded for infinite time $t$, which can be a rather strong assumption and hard to be verified in the real world. This is also related to Equation (13), where we need this equation to be satisfied for infinite time. There are also a lot of assumptions in Theorem 2 that are not justified.

2. The theoretical analysis seems having a gap with the claim. For example, it is claimed that the RL policy trained by the proposed method has UUB stability guarantee, but the RL is actually trained with a reward signal to minimize the error between the reference system and the real system, which is not guaranteed to converge, nor stable.

3. The design of the experiments cannot fully support the claims of the paper. The main claim of the paper is that the learned RL policy has stability guarantees, but the experiments only report the rewards, which is not equivalent to stability.

4. The limitation of the proposed algorithm is not discussed.

5. Although the writing of the paper is mostly clear, the technical parts can be improved. For example, in the first paragraph of page 3, I wonder if $a_1$, $a_2$ are the functions introduced or $a_1(\|\|x\|\|)$ and $a_2(\|\|x\|\|)$. The input of the function should be clear. In the definition of Input to State Stability, $\mathcal{K}$ and $\mathcal{KL}$ are not defined. In the third line of Theorem 2, there is a typo in the equation.

**Questions:**

1. In the abstract, it is said that "current RL-based control systems cannot utilize advanced RL methods while considering stability guarantees". What are "advanced RL methods" here? As far as I know, work like [1], [2] also considers the stability in RL.

2. Is it realistic to assume the state fitting error between the stable reference system and the real system is bounded? Especially for infinite time horizon?

3. It is claimed that if the fitting error between the real system and the reference system converges to zero, the real system can be stable. However, how do you verify this? Can the error go to exact zero in your experiments? Also I think we need to verify this condition for all the states? How do you do that?

4. In Equation 10, $\sigma$ is defined with $i$. What is the difference between each $\sigma_i$?

5. In Theorem 2, there are a lot of assumptions introduced. Can the authors justify them?

6. In the experiments, the Hopper environment is a hybrid system, does this violates the assumption on the continuous system?

7. In the experiments, can the authors verify the learned RL policy is stable? I think high rewards is not equivalent to stability.

8. What are the dimensions of the environments considered in the experiments? Can the algorithm be applied to high dimensional environments?

9. How many random seeds do the authors use for training? Can the authors include standard deviations in the plots?

[1] Chang, Ya-Chien, and Sicun Gao. "Stabilizing neural control using self-learned almost lyapunov critics." 2021 IEEE International Conference on Robotics and Automation (ICRA). IEEE, 2021.

[2] Han, Minghao, et al. "Actor-critic reinforcement learning for control with stability guarantee." IEEE Robotics and Automation Letters 5.4 (2020): 6217-6224.

---

### Official Review · Reviewer_8Gbr · 2023-10-30

**Soundness:** 2 fair
**Presentation:** 2 fair
**Contribution:** 2 fair
**Rating:** 3
**Confidence:** 4

**Summary:**

This paper studies how to use reinforcement learning in control with stability guarantees. It basically trains a controller for a stable system, then if the error between the learned system and the real system is small, the controller can be used for the real system.

**Strengths:**

+ Guaranteeing stability is an important question when RL is used.
+ The results in the paper seem correct.
+ As long as the assumptions in the paper holds, the method can be applied to a broad class of problems.

**Weaknesses:**

- To me the main results from the paper seem too much in the line of "proof-by-assumption". If the error is bounded, and one of the system is stable ($\hat{f}$), then results such as input stability would follow fairly directly?
- Is the bounded error easy to numerically verify? Since the system is nonlinear, then it is possible that the error stays small for some $t$, but one system is stable and the other is unstable.
- The Lyapunov construction seem to be standard.
- Simulations are a bit too simple. It would be good to see how the algorithm works for higher dimensional systems.

**Questions:**

- What happens when the system being controlled is open loop unstable? Would this method learn a controller, say even for a linear system?

---

### Meta-Review · Area_Chair_yhpr · 2023-12-05

**Metareview:**

*Summary*: This paper studies how to use reinforcement learning in control with stability guarantees. It basically trains a controller for a stable system, then if the error between the learned system and the real system is small, the controller can be used for the real system.

*Strengths*: (1) Good writing. (2) Important problem.

*Weaknesses*: (1) Assumptions are too strong and not justified. (2) Theoretical analysis is fairly standard. (3) Experiments are too simple and not conclusive. (4) Missing related work.

*Recommendation*: Reject

**Justification For Why Not Higher Score:**

See the weakness part.

**Justification For Why Not Lower Score:**

See the strengths part.

---

### Decision · Program_Chairs · 2024-01-16

Reject